# Nucleoside Analogues Are Potent Inducers of Pol V-mediated Mutagenesis

**DOI:** 10.3390/biom11060843

**Published:** 2021-06-05

**Authors:** Balagra Kasim Sumabe, Synnøve Brandt Ræder, Lisa Marie Røst, Animesh Sharma, Eric S. Donkor, Lydia Mosi, Samuel Duodu, Per Bruheim, Marit Otterlei

**Affiliations:** 1Department of Clinical and Molecular Medicine, Faculty of Medicine and Health Sciences, NTNU, Norwegian University of Science and Technology, NO-7489 Trondheim, Norway; bksumabe@st.ug.edu.gh (B.K.S.); synnove.b.rader@ntnu.no (S.B.R.); 2West African Centre for Cell Biology of Infectious Pathogens (WACCBIP), University of Ghana, P.O. BOX LG 54 Accra, Ghana; lmosi@ug.edu.gh (L.M.); sduodu@ug.edu.gh (S.D.); 3Department of Biochemistry, Cell and Molecular Biology, University of Ghana, P.O. BOX LG 54 Accra, Ghana; 4Department of Biotechnology and Food Science, Faculty of Natural Sciences, NTNU Norwegian University of Science and Technology, NO-7481 Trondheim, Norway; lisa.m.rost@ntnu.no (L.M.R.); per.bruheim@ntnu.no (P.B.); 5Proteomics and Modomics Experimental Core Facility (PROMEC), NTNU Norwegian University of Science and Technology, NO-7481 Trondheim, Norway; animesh.sharma@ntnu.no; 6Department of Medical Microbiology, University of Ghana Medical School, P.O. Box 4236 Accra, Ghana; esampane-donkor@ug.edu.gh; 7Clinic of Laboratory medicine, St. Olav University Hospital, NO-7006 Trondheim, Norway

**Keywords:** NA, NRTIs, β-clamp, SOS, Pol V, TLS, AMR, MDR

## Abstract

Drugs targeting DNA and RNA in mammalian cells or viruses can also affect bacteria present in the host and thereby induce the bacterial SOS system. This has the potential to increase mutagenesis and the development of antimicrobial resistance (AMR). Here, we have examined nucleoside analogues (NAs) commonly used in anti-viral and anti-cancer therapies for potential effects on mutagenesis in *Escherichia coli*, using the rifampicin mutagenicity assay. To further explore the mode of action of the NAs, we applied *E. coli* deletion mutants, a peptide inhibiting Pol V (APIM-peptide) and metabolome and proteome analyses. Five out of the thirteen NAs examined, including three nucleoside reverse transcriptase inhibitors (NRTIs) and two anti-cancer drugs, increased the mutation frequency in *E. coli* by more than 25-fold at doses that were within reported plasma concentration range (Pl.CR), but that did not affect bacterial growth. We show that the SOS response is induced and that the increase in mutation frequency is mediated by the TLS polymerase Pol V. Quantitative mass spectrometry-based metabolite profiling did not reveal large changes in nucleoside phosphate or other central carbon metabolite pools, which suggests that the SOS induction is an effect of increased replicative stress. Our results suggest that NAs/NRTIs can contribute to the development of AMR and that drugs inhibiting Pol V can reverse this mutagenesis.

## 1. Introduction

Antimicrobial resistance (AMR) is on the rise, posing serious global health threats, including high medical costs and mortality rates. In the United States, AMR is reported to account for nearly 3 million infections and 35 thousand deaths yearly [1]. In Europe, a higher prevalence of AMR infections is reported by countries in the southern and eastern parts of the continent, compared to countries in the north [2]. Exact data are not available for Africa because 40% of the countries do not have records on the incidence of AMR, but the fraction of infections with AMR is increasing [3]. Several Asian countries have a very high incidence of AMR [3] and, in the next 30 years, antibiotic resistance is estimated to cause an annual global mortality rate of 10 million deaths and a USD 100 trillion loss in the global economy, if not addressed. Most of these deaths and losses in the economy are predicted to be from Africa and Asia [4]. Following the outbreak of COVID-19 [5], the increased usage of antibiotics could further accelerate the development of antibiotic resistance [5,6,7].

Bacteria gain resistance to antibiotics via the development of endogenous mutations and/or transfer of resistance genes. Cellular stress, induced, for example, by antibiotics, activates SOS-dependent DNA translesion synthesis (TLS) and this is a major mechanism through which endogenous mutations lead to antibiotic resistance [8]. The ability of the TLS polymerases to introduce mutations is linked to their more spacious catalytic site, enabling a bypass of DNA lesions, and their lack of 3′-5’ exonuclease activity (proofreading activity). There are three TLS polymerases in *Escherichia coli*, Pol II, Pol IV and Pol V (gene product of *polB*, *dinB* and *umuDC*, respectively). In contrast to Pol II and Pol IV, which are also expressed at low levels in absence of SOS response, Pol V is strictly SOS-regulated and, therefore, vital for bacterial mutagenesis [9]. All three bacterial TLS polymerases have been implicated in the generation of mutations contributing to the development of resistance and virulence in bacteria. For example, all three polymerases contribute to the development of mutations conferring ciprofloxacin and rifampicin resistance [8] and both Pol IV and Pol V generate base substitution mutations in the *ampD* gene, conferring resistance to ampicillin [10]. Pol IV is also shown to be relevant for virulence in uropathogenic *E. coli* (UPEC) strains in infected mice [11]. All bacterial polymerases interact with a subunit of the DNA polymerase III (Pol III) holoenzyme, the β-clamp, a homodimer with structural similarities to the mammalian DNA sliding clamp, PCNA (proliferating cell nuclear antigen) [12].

Infections with resistant bacteria will certainly be more devastating in immunocompromised individuals, such as those having HIV/AIDS and cancer, compared to the normal population. Thus, with limited help from the immune system, efficient antibiotics become vital. It is therefore worrying that an increased number of multidrug-resistant (MDR) bacteria have been isolated from HIV-positive individuals compared to HIV-negative individuals in several studies [13,14,15,16]. Characterization of MDR bacteria isolated from HIV patients suggests that endogenous mutations contribute more to the development of resistance than uptake of plasmids carrying drug resistance [17].

Nucleoside analogues (NAs) are used not only in anti-cancer therapies, but also in anti-retroviral therapies as they effectively reduce viral load, and thereby improve the health of the patients and reduce the risk of viral transmission [18]. NAs mimic the natural building blocks of DNA, deoxyribonucleotide triphosphates (dNTPs), and are incorporated into DNA. However, elongations and/or base pairing are blocked by these drugs and this leads to termination of replication [19]. Some NAs, e.g., zidovudine, have a much higher affinity for viral reverse transcriptase than for human polymerases and these are therefore defined as nucleoside reverse transcriptase inhibitors (NRTI) [20].

Here, we have investigated the mutagenic activities of some NAs commonly used in anti-viral and anti-cancer therapies on *E. coli* MG1655. We show that NAs, particularly stavudine, didanosine, 5-fluoro-uracil (5-FU), emtricitabine and fludarabine, increase the mutation frequency by 50–100-fold in *E. coli*. Deletion mutant analysis showed that the increased mutagenesis was driven by the TLS polymerase Pol V, suggesting that the SOS response is induced in the bacteria upon NA treatment. Furthermore, the cell-penetrating APIM-peptide, previously shown to target the polymerase interaction site on the β-clamp and inhibit Pol V, significantly inhibited the NA-induced increase in mutation frequency. This supports a Pol V-dependent mutagenesis. Induction of SOS by stavudine was verified by proteomic analysis and metabolite profiling and showed that SOS was not induced by nucleotide starvation. Data presented here suggest that stavudine induced SOS by interfering with the bacterial replications machinery at doses far below the minimal inhibitory concentration (MIC).

## 2. Materials and Methods

### 2.1. Bacterial Species, Strains and Growth Measurements

*E. coli* K12 (MG1655) wild type (WT) and TLS polymerase deletion strains Δ*umuDC* (ΔPol V) from the Keio collection and Δ*polB*/Δ*dinB* (ΔPol II/ΔPol IV) made in-house as previously described [21] were used. Briefly, *polB* was inactivated in *E. coli* MG1655 ΔPol IV using JW0059: polB. A phage stock (T4GT7) diluted in T4 buffer was added to the JW0059::*polB* culture, incubated and plated on LB agar plates. The lysate was harvested and added to 2 mL T4 buffer and 0.2 mL CHCl3 (T4GT7 lysate). The mixture was centrifuged and supernatant stored at 4 °C, over 20 μL CHCl3. An amount of 0.5 mL overnight culture of the recipient cells was re-suspended in 1 mL of T4 buffer and aliquoted. The T4GT7 lysate (10^−1^, 10^−2^, 10^−3^ and 10^−4^ dilutions) was added to the aliquots, incubated for 25 minutes at room temperature and plated on LB-agar plates with kanamycin. Colonies obtained were re-streaked twice to obtain a pure culture. Kanamycin resistance cassettes replacing the deleted *polB* genes were removed by an FLP-recombinase expressed by pCP20 plasmid.

### 2.2. Peptide

A cell-penetrating peptide targeting the β-clamp named APIM-peptide (Ac-MDRWLVK-GILQWRKI-RRRRRRRR-NH2) (Innovagen, Lund, Sweden) was used. This is the same peptide as RWLVK*, used in Nedal et al. [21], except it has three arginine residues less in its cell-penetrating tail.

### 2.3. Nucleoside Analogues (NAs)

NAs used in this study (listed in Table 1) were purchased from Glentham Life Sciences Ltd. (Wiltshire, UK).

### 2.4. Bacterial Cultivation

Luria Bertani (LB) broth Miller/Microbiological agar and Mueller Hinton broth/agar were used for bacterial cultivation in the rifampicin mutagenicity (Rif^R^) and MIC assays, respectively. Bacterial cultures were incubated at 37 °C, with or without shaking at 250 rpm. Rifampicin (Tokyo chemicals industry Co Ltd., Tokyo, Japan) was used for the Rif^R^ assay.

### 2.5. MIC Assay

The MIC of the nucleoside analogues and APIM-peptide were determined following the protocol used for microtiter broth microdilution assays established by the clinical and laboratory standard institutes [22] with modifications as described [21]. MIC was determined after 24 h.

### 2.6. Rifampicin Mutagenicity Assay (Rif^R^) Assay

This assay was conducted as described [21]. Briefly, pre-cultures of *E. coli* MG1655 (WT and mutant strains) were inoculated in LB broth (Miller) and grown at 37 °C, for 16 h. The pre-cultures were diluted in fresh LB broth (1:100) and 10 mL cultures were grown in 50 mL conical tubes for one hour at 37 °C (shaking incubator at 250 rpm), before they were treated with NAs and/or APIM-peptide. Untreated cultures were used as control. The cultures were further incubated for two hours at 37 °C, before the cultures were serially diluted in LB and plated on LB agar. Undiluted cultures (800 μL) were plated in LB soft agar containing rifampicin (100 µg/mL) on LB agar plates and incubated for 48 h, at 37 °C. The bacterial colony-forming units per millilitre (CFU/mL) were determined from plates with and without rifampicin and the frequency of rifampicin resistance per 10^8^ CFU (Rif^R^/10^8^) was calculated.

### 2.7. Protein Extraction

When harvesting cells for the Rif^R^ assay, 2 × 2 mL of untreated cultures and culture treated with stavudine (Stav) (0.32 µg/mL), APIM-peptide (26 µg/mL), or the combination were pelleted for cell extracts (4000 rpm, 10 min) for each biological replica (*n* = 3). The pellets were resuspended in an OmniCleave™ storage buffer (Lucigen, Middleton, WI, USA) and incubated at room temperature in the presence of 200 U OmniCleave™ endonuclease (Lucigen), 20 µg/µL RNAse (Sigma-Aldrich, St. Louis, MO, USA), 10 U DNAse I (Sigma-Aldrich), 1× phosphatase inhibitor cocktails II and III (Sigma-Aldrich, St. Louis, MO, USA) and 1× cOmplete™ protease inhibitor cocktail (Roche Diagnostics GmbH, Basel, Switzerland). The cells were sonicated on ice at 1.5 output control and 15% duty cycle, for 30 s. The sonication was repeated until a clear phase was observed and cell debris was removed by centrifugation (1400 rpm, 15 min). The supernatants were snap-frozen and stored at −80 °C. The total protein concentration was measured using NanoDrop One C (Thermo Scientific, Waltham, MA, USA) and the Bio-Rad protein assay with a UV-1700 visible spectrophotometer (Pharmaspec, Redmond, WA, USA, 595 nm).

### 2.8. Multiplexed Inhibitor Beads (MIB) Assay and Mass Spectrometry (MS) Analysis of Proteome

Before MS analysis of the protein extract, kinases and other ATP/GTP binding proteins were enriched using the MIB assay as described [23]. A total of 100 µg of the protein extract was analysed per sample.

### 2.9. Metabolite Profiling

When harvesting cells for the Rif^R^ assay, 4 × 4 mL of untreated culture and culture treated with stavudine (Stav) (0.32 µg/mL), APIM-peptide (26 µg/mL), or the combination were also harvested for MS -based metabolite profiling of central carbon metabolite pools, including glycolytic, pentose phosphate pathway (PPP) and tricarboxylic acid cycle (TCA) intermediates, and the complete nucleoside phosphate pool. Complete sampling, processing and MS-based metabolite profiling protocols can be found in previous publications [24,25]. In short, cells were harvested by fast filtration using a vacuum filtering unit with controlled pressure 800 psi below the ambient pressure. After a quick water rinse, filters were transferred to tubes containing 50% ice-cold acetonitrile (VWR chemicals, city, France) for quenching of metabolism. The tubes were immediately snap-frozen and stored at −80 °C. Metabolite extraction was performed through three cycles of freeze-thaw, followed by centrifugation to remove cell debris before the freeze-drying of the cell-free metabolite containing supernatant. Dried supernatants were reconstituted in distilled water, filtered through 3 kDa spin filters (VWR) and analysed on a capillary ion chromatograph (capIC, Thermo Scientific) coupled to a TQ-XS triple quadrupole MS (Waters Corporation, Milford, PA, USA). Samples and analytical grade standards (Sigma-Aldrich, St. Louis, MO, USA) were spiked with yeast extract cultivated with U^13^C-glucose (Sigma-Aldrich, St. Louis, MO, USA) as the sole carbon source, as described in [25].

### 2.10. Data Analysis

Data from the MIB assay were analysed as described [23]. CapIC-MS/MS data were processed in the TargetLynx application manager of MassLynx 4.1 (Waters Corporation, Milford, PA, USA). Absolute quantification was performed by interpolation of calibration curves prepared from serial dilutions of corresponding analytical grade standards (Sigma-Aldrich), calculated by least squares regression with 1/x weighting. Response factors of the analytical standard and biological extracts were corrected by the corresponding response factor of the U^13^C-labeled isotopologue spiked into the samples, a strategy enabling the highest quantitative accuracy and precision [26,27]. Principal component analysis (PCA) was performed on auto-scaled metabolite concentrations normalized to the sum, applying MetaboAnalyst 4.0 [28]. Missing values were replaced by the mean concentration of each metabolite. Unpaired, two-tailed Student’s *t*-tests were used for comparing mutation frequencies; significance levels are described in figure legends.

### 2.11. Mass Spectrometry Data Analysis

Proteins were quantified by processing MS data using MaxQuant v.1.6.17.0 [29]. Open workflow [30] provided in FragPipe version 14 was used to inspect the raw files to determine optimal search criteria. Namely, the following search parameters were used: enzyme specified as trypsin with a maximum of two missed cleavages allowed; acetylation of protein N-terminal, oxidation of methionine, deamidation of asparagine/glutamine and phosphorylation of serine/threonine/tyrosine as dynamic post-translational modification. These were imported in MaxQuant, which uses m/z and retention time (RT) values to align each run against each other sample with 1 min window match-between-run function and 20 min overall sliding window, using a clustering-based technique. These were further queried against the *E. coli* (strain K12) proteome including isoforms downloaded from Uniprot (https://www.uniprot.org/proteomes/UP000000625; accessed on 24 August 2020) and MaxQuant’s internal contaminants database using Andromeda built into MaxQuant. Both protein and peptide identifications false discovery rate (FDR) was set to 1%; only unique peptides with high confidence were used for final protein group identification. Peak abundances were extracted by integrating the area under the peak curve. Each protein group abundance was normalized by the total abundance of all identified peptides for each run and protein by calculated median summing all unique and razor peptide-ion abundances for each protein, using label-free quantification (LFQ) algorithm [31] with minimum peptides ≥1. LFQ values for all samples were combined and log-transformed with base 2 and the transformed control values were subtracted. The resulting values reflecting the change relative to control for each condition were subjected to a two-sided non-parametric Wilcoxon signed rank test [32], as implemented in MATLAB R2020a (Math Works Inc., https://www.mathworks.com/, accessed on 18 March 2020), in order to check the consistency in the directionality of the change, namely, a negative sign reflecting decreased and a positive sign reflecting the increased expression of respective protein groups. The choice of this non-parametric test avoids the assumption of a certain type of null distribution as in the Student’s *t*-test by working over the Rank of the observation instead of the observation value itself. Further, it also makes it robust to outliers and extreme variations noticed in observed values. Differentially expressed (DE) protein groups were identified at *p* 0.25. The UniProt accession IDs of these DE were mapped to pathways (www.wikipathways.org/index.php/DownloadPathways version wikipathways-20201010-gmt-Homo_sapiens.gmt) using R (https://www.R-project.org/, accessed on 19 October 2020) libraries, org.Hs.eg.db and clusterProfiler (www.liebertpub.com/doi/10.1089/omi.2011.0118). Venn diagrams were built using the R package limma [33] and heatmap using pheatmap (https://cran.r-project.org/web/packages/pheatmap/index.html, accessed on 4 January 2020). The raw data with results have been deposited in the ProteomeXchange Consortium via the PRIDE [34] partner repository with the dataset identifier, project ID PXD025370 (Reviewer account Username: reviewer_pxd025370@ebi.ac.uk Password: seTUpXp4).

## 3. Results and Discussion

### 3.1. NAs Induce Mutagenesis in Bacteria at Concentrations That Do Not Affect Bacterial Growth

Antibacterial activities of some anti-viral and anti-cancer NAs have been reported previously [19]. Since doses below the MIC of the different NAs must be used to test the mutagenic potential of these drugs, we first determined the MICs of the different NAs on *E. coli* MG1655. Among the NAs tested (Table 1), only stavudine, didanosine and 5-FU inhibited visible bacterial growth at concentrations below 256 µg/mL (maximal concentration examined). The lack of antibacterial activities of several of these NAs, e.g., cytarabine, vidarabine, fludarabine, abacavir, emtricitabine, lamivudine and tenofovir, are in agreement with published data [35,36].

To examine the mutagenic potential of the different NAs, *E. coli* MG1655 was treated with 0.5× MIC or 0.25× of the maximal concentration used in the MIC assay (256 µg/mL). The NRTIs stavudine, didanosine and emtricitabine induced a strong (45–125×) increase in mutation frequency (Figure 1A). Abacavir and lamivudine, two other NRTIs, also significantly increased the mutation frequency (~10–15×). 5-FU and fludarabine, two NAs commonly used in anticancer therapy, increased the mutation frequency by more than 50 times. The sub-MIC doses used in Figure 1A are in all cases except for 5-FU higher than plasma concentrations obtained in patients (see Table 1). We therefore examined doses that are in the human plasma concentrations range (Pl.CR) (Figure 1B) and found that in addition to 5-FU, stavudine, didanosine, emtricitabine and fludarabine were potent inducers of mutagenesis at clinically relevant doses. Thus, these drugs may increase bacterial mutagenesis and contribute to hard-to-treat bacterial infections when used clinically. It still remains elusive why these NAs are more mutagenic than the other NAs tested. Stavudine was the most potent inducer of mutagenesis. A more than 50× increase in mutation frequency was detected at 0.016 µg/mL, a dose 20-fold lower than the reported upper levels of Pl. CR (0.32 µg/mL). That NAs increase the mutation frequency in bacteria may, at least partly, explain why a high fraction of antibiotic-resistant bacteria is isolated from HIV-positive patients [13,14,15,16,35].

### 3.2. Pol V Is the TLS Polymerase Responsible for the Mutagenesis Induced by NAs

To determine which TLS polymerase is responsible for the induced mutagenesis, the mutation frequency in WT, a strain with deletion of Pol V (ΔPol V) and a Pol IV and Pol II double deletion strain (ΔPol II/ΔPol IV) of *E. coli* MG1655 were examined after treatment with stavudine. The results clearly indicate that the main TLS polymerase responsible for the increase in mutation frequency is Pol V and that the deletion of both the other two TLS polymerases in *E. coli.,* Pol IV and Pol II, did not significantly reduce the mutation frequency (Figure 2A). Pol V is considered to be the main SOS inducible TLS polymerase in bacteria (reviewed in [51]).

We have previously shown that peptides containing the PCNA interacting motif APIM, bind to the β-clamp and block the function of Pol V [21]. Here we show that the APIM-peptide was able to reduce the mutation frequency induced by stavudine when these were combined (combination) (Figure 2B), further supporting a Pol V-dependent mutagenesis.

### 3.3. Stavudine Treatment Induces only Minor Changes in Central Carbon Metabolite Pools

Stavudine is a thymidine nucleoside analogue, which when incorporated into DNA by reverse transcriptase, terminates the replication of the virus. Pyrimidine synthesis is strictly regulated and some NAs, such as 5-FU, are also thymidylate synthase inhibitors. Thus, NAs may affect the nucleotide pools and this may both directly and indirectly stress the cells and increase the mutation frequency. To explore the effects of stavudine on the nucleotide pools, quantitative MS-based metabolite profiling of *E. coli* treated with stavudine and APIM-peptide alone and the combination of stavudine and APIM-peptide (combination) were performed. PCA did not reveal significant differences in the metabolite pool composition of the treated cells and the untreated control (95% confidence interval, Figure 3A). However, the combination treatment clustered closer to the APIM-peptide single treatment than the stavudine single treatment (Figure 3A), indicating a larger similarity between the two. This is also evident from inspecting fold changes on the level of individual metabolites; the central carbon metabolite pool in APIM-peptide and combination-treated *E. coli* were more similar and, more specifically, more down-regulated, compared to untreated control than stavudine single treatment (Figure 3B). Especially noteworthy is the slight increase in pyrimidine nucleoside pools in stavudine single treated cells, as opposed to down-regulated pools in both in the APIM-peptide- and the combination-treated group. Down-regulation of nucleoside pools after treatment with APIM-peptide has also been also detected in mammalian cancer cells [52]. However, the increased mutation frequency observed by stavudine as a single agent cannot be explained by cellular stress induced by the small increase in nucleotide pools detected.

### 3.4. Stavudine Treatment Activates the SOS System

When harvesting samples for mutation frequency and metabolite profiling, cells were also harvested for proteome analysis using the MIB assay [23]. This is an assay where activated signalling proteins, and proteins binding to these, are pulled down via their affinity for ATP/GTP using kinase inhibitors (also NAs) immobilized on sepharose beads. The proteins attached are trypsinized and identified by MS/MS. Approximately 1000 proteins were pulled down from each sample. During analysis, only proteins that were similarly changed relative to untreated control in all three biological replicas (significant change, Wilcoxon test [23]) were included in downstream analysis. More proteins were similarly changed in *E. coli* treated with stavudine and the combination, than in APIM-peptide-treated *E. coli* compared to either of the two groups (Figure 4A). This suggests that the APIM-peptide induces a different response than stavudine. When examining the change in proteins known to be related to the bacterial SOS response [53], more SOS proteins were found to change in stavudine and combination-treated cells, than in the APIM-peptide-treated cells (Figure 4B). Heatmaps of the log_2_ fold change in pull-down of proteins found to be significantly changed in at least one treatment group showed that most of the changed SOS regulated proteins were increased relative to control (Figure 4C). This was expected as the mutation frequencies increased. Interestingly, RecA was increased in both stavudine and the combination, but not in APIM-peptide-treated cells. The opposite was the case for LexA, which increased significantly only in APIM-peptide-treated cells. A trend towards a reduction in LexA was detected in stavudine and combination-treated bacteria, albeit not significant. These data support that stavudine is inducing an SOS response in bacteria at doses far below MIC. APIM-peptide is not inducing an SOS response, nor reducing the SOS response induced by stavudine when used in combination. This was expected as the APIM-peptide is believed to inhibit the interactions between the β-clamp and Pol V and thereby inhibit the increase in mutation frequency, but not to inhibit the SOS induction [21].

We next examined which pathways were changed upon treatment using information from stress responses previously described in ciprofloxacin-treated bacteria [54]. We found that stavudine treatment led to a reduced pull-down of several proteins involved in general metabolism, while APIM-peptide treatment led to an increased pull-down of more proteins involved in nucleotide metabolism (Figure 4D). The latter result is interesting as APIM-peptide also affected the nucleotide pools more than stavudine.

In summary, the most likely explanation for the strong increase in mutation frequency is that stavudine induces SOS at doses below MIC, likely via interference with normal bacterial replications leading to replication stress.

## 4. Conclusions

Our findings presented herein indicate that anti-viral and anti-cancer NAs are capable of inducing bacterial mutagenesis and contributing to high AMR in HIV and cancer patients at clinically relevant concentrations. Stavudine induces mutagenesis by activating the *E. coli* SOS system and, thereby, the TLS polymerase Pol V. Thus, the combination of NAs/NRTIs and antibiotic treatment in already vulnerable patients may drive the development of and selection for, antibiotic-resistant bacteria. APIM-peptide treatment significantly reduced the stavudine-induced mutagenesis suggesting that this peptide could be a good drug candidate in the fight against AMR.

## Figures and Tables

**Figure 1 biomolecules-11-00843-f001:**
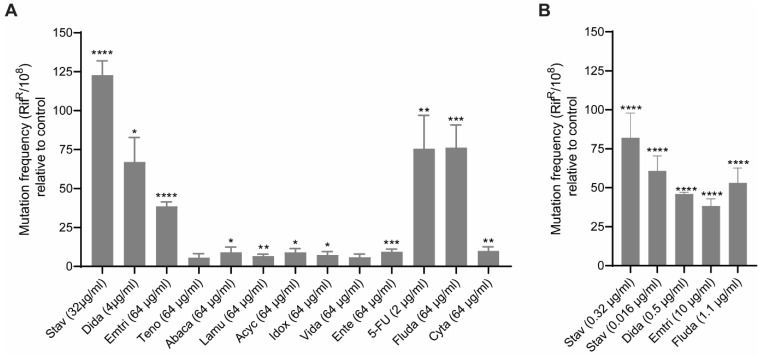
Nucleoside analogues (NAs) induce mutagenesis in *Escherichia coli* MG1655**.** Data represents the average increase in mutation frequency in *E. coli* MG1655 cells detected in the Rif^R^ assay after treatment, compared to untreated control. (**A**) Mutation frequencies determined using 0.5× MIC or 0.25× of maximal examined concentration (256 µg/mL) (Table 1) of the different NAs. The relative increase in mutation frequency is given from 5 independent biological replicas. Stavudine (Stav, 32 µg/mL), didanosine (Dida, 4 µg/mL), emitrcitabine (Emtri, 64 µg/mL), tenofovir (Teno, 64 µg/mL), abacavir (Abaca, 64 µg/mL), lamuvudine (Lamu, 64 µg/mL, acyclovir (Acyc, 64 µg/mL), idocuridine (Idox, 64 µg/mL), vidarabine (Vida, 64 µg/mL), entecavir (Ente, 64 µg/mL), 5-fluro uracil (5-FU, 2 µg/mL), fludarabine, (Fluda, 64 µg/mL) and cytarabine (Cyta, 64 µg/mL). (**B**) Mutation frequency at plasma concentrations of Stav (0.32 µg/mL and 0.016 µg/mL), Dida (0.5 µg/mL), Emtri (10 µg/mL) and Fluda (1.1 µg/mL). The relative increase in mutation frequency is given from a minimum of 3 independent biological replicas, except for 0.32 µg/mL Stav, which are from 14 independent biological replicas. Each biological replica consists of 2–3 technical replicates. Unpaired Student’s *t*-test, two-tailed, **** *p* < 0.0001, *** *p* < 0.001, ** *p* < 0.01 and * *p* < 0.05.

**Figure 2 biomolecules-11-00843-f002:**
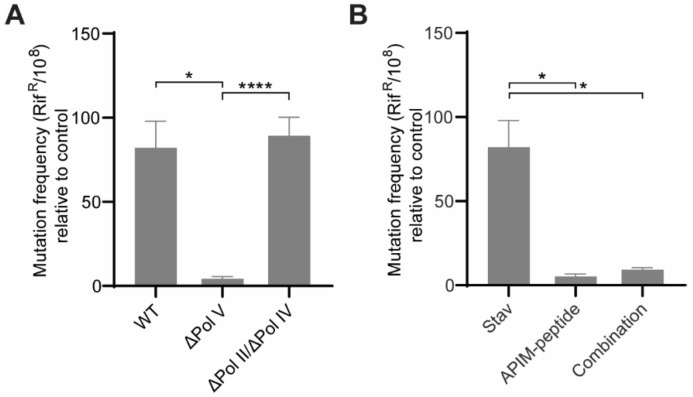
Stavudine-induced mutagenesis in *Escherichia coli* MG1655 is driven by the translesion synthesis (TLS) Pol V. (**A**) Data represent the average increase in mutation frequency using the Rif^R^ assay after treatment of stavudine (Stav, 0.32 µg/mL), compared to untreated control for *E. coli* MG1655 wild type (WT) and TLS polymerase deletion strains (ΔPol V and ΔPol II/Pol IV). (**B**) Data represent the average increase in mutation frequency using the Rif^R^ assay after treatment of stavudine (Stav, 0.32 µg/mL), APIM-peptide (26 µg/mL) and a combination (Stav + APIM-peptide), compared to untreated control for *E. coli* MG1655 wild type (WT). The mutation frequency from the stavudine-treated WT cells (WT and Stav) are from 14 independent biological replicas. The mutation frequencies from the remaining samples (ΔPol V and ΔPol II/Pol IV, APIM-peptide and combination) are from 5 independent biological replicas, where 3 technical replicates were plated from each biological replicate. Unpaired Student’s *t*-test, two-tailed, **** *p* < 0.0001 and * *p* < 0.05.

**Figure 3 biomolecules-11-00843-f003:**
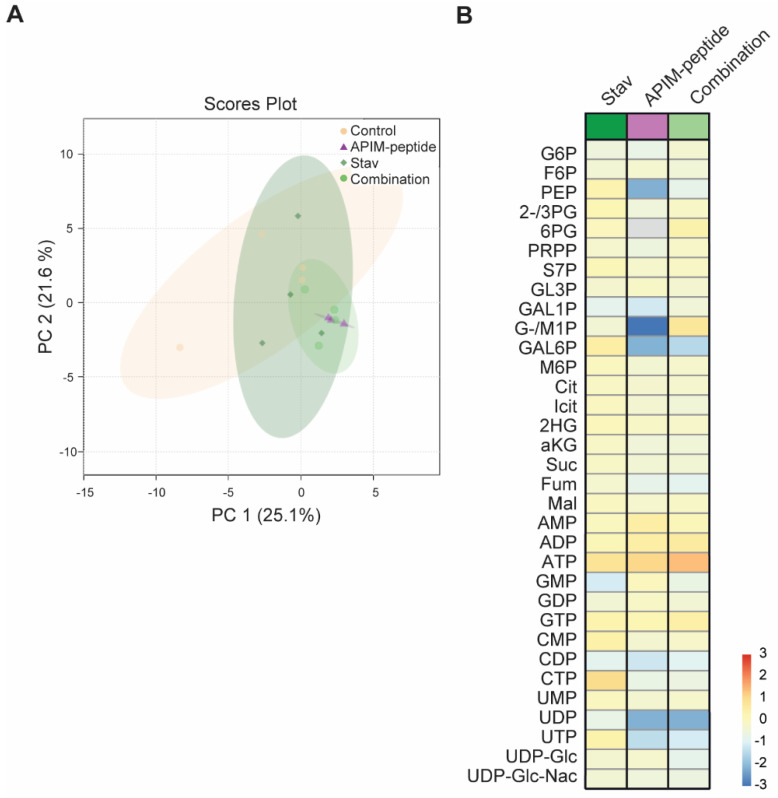
Stavudine treatment of *Escherichia coli* MG1655 leads to small changes in glycolytic, PPP and TCA intermediates and in nucleoside phosphate pools. (**A**) Scores plot (PC1 and PC2) from PCA of intracellular metabolite concentrations, as measured by capIC-MS/MS, for the four treatment groups: stavudine (Stav, 0.32 µg/mL), APIM-peptide (26 µg/mL) and combination (Stav + APIM-peptide) and untreated control harvested two hours after treatment, with 95% confidence regions indicated. (**B**) Log_2_ fold change of treated cultures vs. untreated control. Data are averages of four independent biological replicas normalized by sum.

**Figure 4 biomolecules-11-00843-f004:**
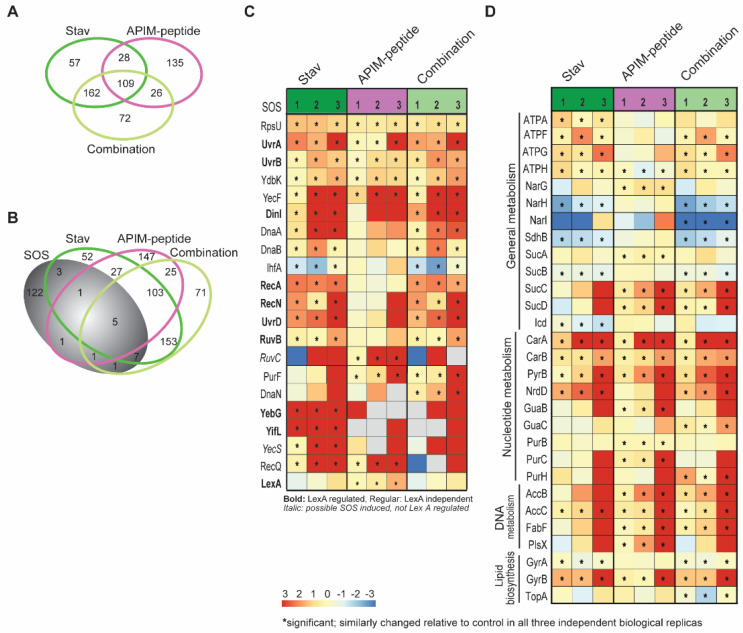
Stavudine induces SOS in *Escherichia coli*. Results of the MIB assay of extract from *E. coli* MG1655 treated with stavudine (Stav, 0.32 µg/mL), APIM-peptide (26 µg/mL) and the combination (Stav + APIM-peptide) for 2 h. Data from three independent biological replicas are shown. Protein concentrations are given as log_2_ fold change relative to untreated control. (**A**) Venn diagrams displaying the number of proteins significantly changed in pull-downs from treated cells relative to untreated control according to the Wilcoxon signed-rank test. (**B**) Venn diagram showing an overlap in the different treatment groups with proteins involved in SOS [53]. (**C**) Heatmaps displaying log_2_ fold change in protein concentration of SOS proteins relative to untreated control in the different treatment groups. The grey boxes represent no data. (**D**) Heatmaps displaying log_2_ fold change in proteins concentration of proteins involved in general metabolism, nucleotide metabolism, lipid biosynthesis and DNA metabolism relative to untreated control in the different treatment groups. * Proteins that are significantly changed according to the Wilcoxon signed-rank test (i.e., changed the same way in all three replicas).

**Table 1 biomolecules-11-00843-t001:** Nucleoside analogues (NAs) and their minimal inhibitory concentrations (MICs) in *Escherichia coli* MG1655.

Nucleoside Analogue (NA)	Reported Plasma Concentration (Pl.CR.) (µg/mL)	MIC (µg/mL)	Structure	Use
Stavudine(Stav)	≤0.9[37]	64	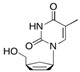	HIV, NRTI. dTTP analogue
Didanosine(Dida)	0.05–0.86[38]	8	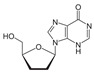	HIV NRTI. dATP analogue
Emtricitabine(Emtri)	1.01–11.5[39]	>256	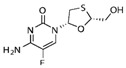	HIV NRTI. dCTP analogue
Tenofovir(Teno)	0.050–0.077[40]	>256	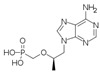	HIV NRTI. dTTP analogue
Abacavir(Abaca)	0.04–1.13[41]	>256	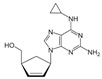	HIV NRTI. dGTP analogue
Lamivudine(Lamu)	0.02–0.11[42]	>256	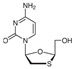	HIV-1 NRTI. dTTP analogue
Acyclovir(Acyc)	10.0[43]	>256	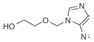	Herpesviruses. dGTP analogue
Idoxuridine(Idox)	10.0–36.0[44]	>256	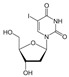	Herpesviruses. dUTP analogue. Thymidylate phosphatase inhibitor.
Vidarabine(Vida)	0.2–0.4[45,46]	>256	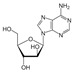	Herpesviruses. dATP analogue
Entecavir(Ente)	0.015–0.036[47]	>256	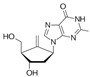	Hepatitis virus. dGTP analogue
5-Fluoro Uracil(5-FU)	0.025–8.0[48]	4	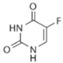	Multiple cancers. Thymidylate synthase inhibitor
Fludarabine(Fluda)	0.4–2.0[49]	>256	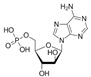	Leukaemia and lymphoma. dATP analogue
Cytarabine(Cyta)	~486.0[50]	>256	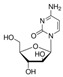	Leukaemia. dCTP analogue

## Data Availability

The raw data is deposited in PRIDE, project ID PXD025370 (Reviewer account Username: reviewer_pxd025370@ebi.ac.uk Password: seTUpXp4).

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
