# Peer review of "Nucleoside Analogues Are Potent Inducers of Pol V-mediated Mutagenesis"

_biomolecules, 2021, doi:10.3390/biom11060843_

Round 1

Reviewer 1 Report

I like this paper. It may be published after minor editing.

Author Response

. We thank the reviewer for the positive feedback, and apologize for the many spelling mistakes. We have now gone through the paper and made necessary corrections.

Reviewer 2 Report

The manuscript addresses an interesting topical issue. Combating AMR is a necessary action with a significant impact in treating microbial infections. The mutagenic activities of some NAs is a valuable study that may be the basis for other studies focused on combating AMR. The argumentation of the results is correct.

The manuscript is well structured and easy to read and understand. Still, the article needs some improvement in terms of content and writing.

Title. The chosen title of the manuscript is far too general. I suggest the authors identify a more specific title according to the proposed objectives and the obtained results.

Abstract. APIM-peptide is missing from the abstract content; thus, APIM-peptide is a crucial element in the Conclusions part.

The Introduction part sets the objectives to be achieved and do not have to present the obtained results. See page 2 (84-96 lines).

The legend of the figures is too long and difficult to follow. Please, insert in the manuscript some details.

I suggest authors use simpler abbreviations for NAs.

Please correct:

  • a missing space „endonuclease(Lucigen)”, on page 3, line 139
  • „anlyzed” on page 4, line 149
  • „ include ingglycolytic” on page 4, line 154
  • „weretransferred” on page 4, line 159
  • „desrcibed” on page 4, line 169
  • „elsusive” on page 5, line 245
  • „stauvidune” on page 8, line 300
  • „inducated” on page 8, line 304
  • „inividual” on page 8, line 305

and so on.

English writing needs to be improved.

Author Response

Point-to-point rebuttal, our response in italic

The manuscript addresses an interesting topical issue. Combating AMR is a necessary action with a significant impact in treating microbial infections. The mutagenic activities of some NAs is a valuable study that may be the basis for other studies focused on combating AMR. The argumentation of the results is correct.

The manuscript is well structured and easy to read and understand. Still, the article needs some improvement in terms of content and writing.

Our response: We thank the reviewer for the positive comments. We have addressed the reviewer’s concerns point-by-point below. 

Title. The chosen title of the manuscript is far too general. I suggest the authors identify a more specific title according to the proposed objectives and the obtained results. We appreciate that the

Our response: We agree that the title is general, and we have now added “Pol V-mediated mutagenesis" to the title to make it more specific to our findings.

Abstract. APIM-peptide is missing from the abstract content; thus, APIM-peptide is a crucial element in the Conclusions part.

Our response: We agree, and have now included this in the abstract.

The Introduction part sets the objectives to be achieved and do not have to present the obtained results. See page 2 (84-96 lines).

Our response: We understand that the referee thinks that a repetition of the results is redundant at the end of the introduction, but we think that summarizing the main result in somewhat more detail than in the abstract is useful for the reader.

The legend of the figures is too long and difficult to follow. Please, insert in the manuscript some details.

Our response: We aimed to make figure legends that included all the data necessary for understanding the figures without having to read the text. Therefore, we would rather not move information from the legend to the text.

I suggest authors use simpler abbreviations for NAs.

Our response: We agree that the abbreviations are long; however, as we only use them in the figures and not in the text, we suggest keeping them like this.

Please correct:

  • a missing space „endonuclease(Lucigen)”, on page 3, line 139

Corrected

  • „anlyzed” on page 4, line 149

Corrected

  • „ include ingglycolytic” on page 4, line 154

Corrected

  • „weretransferred” on page 4, line 159

Corrected

  • „desrcibed” on page 4, line 169

Corrected

  • „elsusive” on page 5, line 245

Corrected

  • „stauvidune” on page 8, line 300

Corrected

  • „inducated” on page 8, line 304

Corrected

  • „inividual” on page 8, line 305

Corrected

and so on.

English writing needs to be improved.

Our response: We apologize for the multiple spelling mistakes. We have now gone through the paper and made necessary corrections.

Reviewer 3 Report

Paper presented by Sumabe et al., describes the usage of nucleoside analogues in the mutagenesis of bacterial cells. I found this text very interesting, it is smooth and comprehensive collaboration on the topic. 

The research design is appropriate, each step is well thought out and forms a coherent whole. Additional study with the SOS response make it valuable research paper. 

  • Please describe method used for preparation of mutants ΔpolB/ΔdinB 
  • Could authors add a short discussion about possible toxicity of NAs/NRTIs to eukaryotic cells? Would it be possible to apply the combination of NAs/NRTIs and antibiotics in clinical settings as another strategy to overcome bacterial resistance?

Author Response

Point-to-point rebuttal, our response in italic

 Paper presented by Sumabe et al., describes the usage of nucleoside analogues in the mutagenesis of bacterial cells. I found this text very interesting, it is smooth and comprehensive collaboration on the topic. 

The research design is appropriate, each step is well thought out and forms a coherent whole. Additional study with the SOS response make it valuable research paper. 

Our response: We thank the reviewer for the positive comments.

  • Please describe method used for preparation of mutants ΔpolB/ΔdinB 

Our response: This is done as described in Nedal et al, NAR,2020. We have now, in addition to referring to this paper, added the brief description in material and methods

  • Could authors add a short discussion about possible toxicity of NAs/NRTIs to eukaryotic cells? Would it be possible to apply the combination of NAs/NRTIs and antibiotics in clinical settings as another strategy to overcome bacterial resistance?

Our response: To combine NAs/NRTIs with antibiotics in clinical settings is a combination that induce mutagenesis and select for resistance, and something that we should avoid. This is addressed in the conclusion “Thus, the combination of NAs/NRTIs and antibiotic treatment in already vulnerable patients may drive development of, and selection for, antibiotic resistant bacteria.” One possibility to avoid this is to combine NAs/NRTIs, antibiotic and an anti-TLS drug, here exemplified by the APIM-peptide, as suggested in the conclusion “APIM-peptide treatment significantly reduced the stavudine-induced mutagenesis suggesting that this peptide could be a good drug candidate in the fight against AMR.”

Reviewer 4 Report

In this research article, the authors examined nucleoside analogues (NAs) commonly used in anti-viral and anti-cancer therapies for potential effects on mutagenesis in Escherichia coli using the Rifampicin mutagenicity assay. This work highlights a very important global health issue. Today, antibiotic resistance is endemic in many countries of the world, while the discovery of new antibiotic compounds is lagging.

This work will be of broad interest to readers of the Biomolecules journal.

Minor comment:

  1. The names of the microorganisms should be written in italic in some cases should be corrected please check the entire manuscript.
  2. Errors in the cited references

Author Response

Point-to-point rebuttal, our response in italic

Suggestions for Authors

In this research article, the authors examined nucleoside analogues (NAs) commonly used in anti-viral and anti-cancer therapies for potential effects on mutagenesis in Escherichia coli using the Rifampicin mutagenicity assay. This work highlights a very important global health issue. Today, antibiotic resistance is endemic in many countries of the world, while the discovery of new antibiotic compounds is lagging.

This work will be of broad interest to readers of the Biomolecules journal.

Our response: We thank the reviewer for the positive comments.

Minor comment:

  1. The names of the microorganisms should be written in italic in some cases should be corrected please check the entire manuscript.

Our response: We apologize for this, and have now done corrections

  1. Errors in the cited references

Our response: We apologize for this, corrections have been made